# LncRNA TUG1/miR-29c-3p/SIRT1 axis regulates endoplasmic reticulum stress-mediated renal epithelial cells injury in diabetic nephropathy model *in vitro*

**Shaoqiang Wang[1], Pengfei Yi[2], Na Wang[2], Min Song[2], Wenhui Li[1], Yingying Zheng[2]\***

**1** Department of Thoracic Surgery, Affiliated Hospital of Jining Medical University, Jining Medical University, Jining city, Shandong Province, P.R.China, **2** Department of Endocrinology, Affiliated Hospital of Jining Medical University, Jining Medical University, Jining city, Shandong Province, P.R.China

\* zyy926@mail.jnmc.edu.cn

**Data Availability Statement:** All relevant data are within the paper and its Supporting information files.

## Abstract

Long non-coding RNAs (lncRNAs) are important regulators in diabetic nephropathy. In this study, we investigated the potential role of lncRNA TUG1 in regulating endoplasmic reticulum stress (ERS)-mediated apoptosis in high glucose induced renal tubular epithelial cells. Human renal tubular epithelial cell line HK-2 was challenged with high glucose following transfection with lncRNA TUG1, miR-29c-3p mimics or inhibitor expression plasmid, either alone or in combination, for different experimental purposes. Potential binding effects between TUG1 and miR-29c-3p, as well as between miR-29c-3p and SIRT1 were verified. High glucose induced apoptosis and ERS in HK-2 cells, and significantly decreased TUG1 expression. Overexpressed TUG1 could prevent high glucose-induced apoptosis and alleviated ERS via negatively regulating miR-29c-3p. In contrast, miR-29c-3p increased HK-2 cells apoptosis and ERS upon high glucose-challenge. SIRT1 was a direct target gene of miR-29c-3p in HK-2 cells, which participated in the effects of miR-29c-3p on HK-2 cells. Mechanistically, TUG1 suppressed the expression of miR-29c-3p, thus counteracting its function in downregulating the level of SIRT1. TUG1 regulates miR-29c-3p/SIRT1 and subsequent ERS to relieve high glucose induced renal epithelial cells injury, and suggests a potential role for TUG1 as a promising diagnostic marker of diabetic nephropathy.

## Introduction

Diabetic nephropathy is one of the most common complications of diabetes and is the major triggering factor for kidney failure. Typical symptoms of diabetic nephropathy include blood pressure dysregulation, loss of appetite, nausea and vomiting, persistent itching and fatigue, causing great physical and mental suffering to diabetic nephropathy patients[1, 2]. Diabetic nephropathy patients often have a high risk of mortality, mostly caused by cardiovascular complications[3]. The development of diabetic nephropathy is a multifactorial process, and emerging evidences have unveiled the crucial role of endoplasmic reticulum stress (ERS) in

**Funding:** This work was supported by the Research Fund for Academician Lin He New Medicine (No. JYHL2019FMS11); Nursery research project of the Affiliated Hospital of Jining Medical University (No. MP-2018-001); Teacher Research Support Fund of Jining Medical University (JY2017FS002); National Natural Science Foundation of China (No. 81802290); Natural Science Foundation of Shandong Province (No. ZR2018BH020).

**Competing interests:** The authors have declared that no competing interests exist.

**Abbreviations:** ANOVA, one-way analysis of variance; Bax, BCL2-associated X protein; Bcl-2, B-cell lymphoma 2; CHOP, C/EBP homologous protein; DMEM, Dulbecco's Modified Eagle Medium; ECL, enhanced chemi-luminescence; eIF-2α, eukaryotic initiation factor 2α; ERS, endoplasmic reticulum stress; GRP78, Glucose-regulated protein 78 kDa; HG, high glucose; lncRNA, long non-coding RNA; miRNA, microRNA; MTT, 3-(4,5-dimethyl-thiazol-2-y1)-2,5-diphenyl tetrazolium bromide; PERK, PKR-like ER kinase; PVDF, polyvinylidenedifluoride; RTECs, renal tubular epithelial cells; SD, standard deviation; SIRT1, sirtuin 1; UPR, unfolded protein response.

regulating the pathological processes of diabetic nephropathy[4, 5]. ERS can be induced by diverse kinds of environmental stresses in the diabetic kidneys such as high level of glucose or free fatty acids. Under these conditions, renal tubular epithelial cells (RTECs) are prone to develop ERS which often lead to the apoptosis of RTECs and the consequent kidney injury[5, 6]. Therefore, it has been well recognized that reducing ERS can prevent RTECs from apoptosis and alleviate the symptoms of various renal disorders, including diabetic nephropathy[7].

Long non-coding RNAs (lncRNAs) are a type of RNA molecules that typically exceed 200 nucleotides in length[8]. Although without protein-coding ability, lncRNAs participate in regulating various kinds of physiological and pathological processes[9, 10]. However, in most cases, the detailed mechanisms underlying their function are poorly understood. Recently, several lncRNAs have been identified to be involved in the development of diabetic nephropathy [11, 12]. For instance, lncRNA Gm4419 enhanced the proinflammatory and proliferative capacities of mesangial cells by augmenting NF-κB/NLRP3 signaling pathway, through which Gm4419 promoted the development of diabetic nephropathy[13]. In contrast, CYP4B1-PS1-001, another lncRNA, could suppress the proliferation of mesangial cells in high glucose conditions[14]. LncRNA TUG1 (taurine up-regulated gene 1), a lncRNA which locates on human chromosome 22q12.2 and plays an oncogenic role in many kinds of human cancers, was reported to alleviate the histological damage of diabetic nephropathy in diabetic mice by enhancing the expression of PGC-1α in podocytes[15]. TUG1 was also reported to suppress the accumulation of extracellular matrix by inhibiting the function of miR-377 in high glucose-treated mesangial cells[16]. These findings suggest the protective role of TUG1 in the development of diabetic nephropathy. However, whether TUG1 is involved in regulating ERS in injured RTECs remains unknown.

Diabetes-related miR-29c-3p is a characteristic miRNA under high glucose conditions and a marker of renal fibrosis, which is involved in inducing apoptosis and increasing the accumulation of extracellular matrix proteins[17]. MiR-29c-3p knockout in vivo prevented the progression of diabetic nephropathy[17]. The application of DIANA-tool and LncBase bioinformatics software prediction and literature reports show that LncRNA TUG1 can be targeted combined with miR-29c-3p[18–20], to speculate that TUG1 has the potential to interfere with diabetic nephropathy.

SIRT1 (NAD-dependent deacetylase sirtuin-1) is abundant in kidney, which is closely related to renal physiology and pathology, and involved in the regulation of diabetic nephropathy[20, 21]. SIRT1 is a key molecule in glucose, lipid and energy metabolism, and plays an important role in protecting renal cells from cellular stress which by participating in the deacetylation of transcription factors such as P53, FOXO, RelA/P65, NFκ-B, STAT3 and PPARγ[21, 22]. SIRT1 regulate ERS through PERK/eIF-2α/CHOP axis[23].

In the present work, we reported a regulatory mechanism of TUG1 in the pathological process of diabetic nephropathy *in vitro*. TUG1 could alleviate high glucose-induced ERS in human RTECs and protected them from apoptosis. Mechanistically, TUG1 supported the expression of SIRT1, a well-characterized deacetylase responsible for regulating cellular glucose metabolism, by down-regulating the level of miR-29c-3p. Thus, our study uncovers a mechanism involving lncRNA TUG1, miR-29c-3p and SIRT1 in regulating ERS-induced cell damage in RTECs.

## Methods

### Cell culture and high glucose challenge

HK-2 cells were purchased from Shanghai Cell Bank of Chinese Academy of Science (Shanghai, China) and was maintained (at 37°C in a humidified atmosphere of 5% $CO_2$) in

Dulbecco's Modified Eagle Medium (DMEM) (Gibco, Los Angeles, CA, USA) supplemented with 10% fetal bovine serum and 1% penicillin-streptomycin. For high glucose challenge, HK-2 cells were cultured in DMEM medium containing various concentrations (15, 30, 45 mM) of D-glucose (Sigma-Aldrich, St. Louis, MO, USA) for 48 h.

## Flow cytometry

Trypsin-digested HK-2 cells were stained with FTIC-Annexin V/PI (Beyotime, Shanghai, China) for 15 min in dark at room temperature followed by addition of 1 ml PBS. Cells were then centrifuged, and cell pellets were resuspended in 100 μl PBS. Flow cytometry was performed on a FACS Calibur (BD Biosciences, USA) to examine the apoptosis of HK-2 cells. All experiments were conducted in triplicate.

## Cell transfection

For overexpression of lncRNA TUG1, TUG1 cDNA was cloned into the pcDNA3.1 vector (Invitrogen, Carlsbad, CA, USA). miR-29c-3p mimics (5′–UAGCACCAUUUGAAAUCGGUUA–3′), miR-29c-3p inhibitor (5′–UAACCGAUUUCAAAUGGUGCUA–3′), negative control mimics (5′–GACCAGAGUCCCGUACUCCU–3′) or negative control inhibitor (5′–AAGGCUAGCAUAGAAUCGUA–3′) oligonucleotide (mimics NC, inhibitor NC) were purchased from RiboBio (Guangzhou, China). MiR-29c-3p mimics are *in vitro*-synthesized RNA oligonucleotides that have the similar sequence to the endogenous miR-29c-3p and enhance its function. While, miR-29c-3p inhibitor can bind to the endogenous miR-29c-3p and neutralize its function. Plasmids were transfected into HK-2 cells using Lipofectamine 2000 Reagent (Life Technologies, Carlsbad, CA, USA) following manufacturer's instructions. After 48h incubation, cells were treated with high glucose for further experiments.

## Real-time PCR

Total RNA of HK-2 cells was extracted using Trizol reagent followed by reverse-transcribed into cDNA using PrimerScript™ RT Master Mix (Takara, Shiga, Japan). Real-time PCR reaction was performed on an ABI 7500 Real-Time PCR System (Applied Biosystems, USA). The relative mRNA expression of TUG1 genes were calculated using $2^{-\Delta\Delta Ct}$ method. lncRNA TUG1 or miR-29c-3plevel was normalized to U6, the expression of mRNA was normalized to GAPDH.

## Western blot

The lysates of HK-2 cells were prepared with RIPA lysis buffer containing Protease and Phosphatase Inhibitor Cocktail (Cell Signaling Technology, Boston, USA). Then the concentrations of total protein were measured by BCA method (Thermo Fisher Scientific, USA). 30 μg protein were subjected to SDS-PAGE (10%), and then transferred to polyvinylidenedifluoride (PVDF) membranes (Millipore, Beford, MA, USA). After blocking with5% bovine serum albumin, the membranes were incubated with appropriate primary antibodies at 4˚C overnight, followed by incubation with HRP-conjugated secondary antibody (1:5000, Beyotime, Shanghai, China) at room temperature for 1 h. Following antibodies were used: anti-PERK (1:1000), anti-p-PERK (1:1000), anti-GPR78 (1:1000), anti-CHOP (1:1000), anti-cleaved Caspase12 (1:1000), anti-cleaved Caspase3 (1:1000) (Cell Signaling Technology, USA), anti-GAPDH (1:3000), anti-p-eIF-2α (1:1000), anti-eIF-2α (1:1000), anti-Bax (1:1000), anti-Bcl-2 (1:1000) (Abcam, Cambridge, MA, USA). GAPDH was selected as the loading control. The blots were detected using the enhanced chemi-luminescence (ECL) detection kit (KeyGen Biotech, Nanjing, China).

### Dual-luciferase reporter assay

Dual-luciferase reporter assay was performed as previously described[24] using Dual-Luciferase Reporter kit (Promega, Madison, WI, USA) following the manufacturer's instruction. The amplified sequences (TUG1, miR-29c-3p and SIRT1) were cloned into pmirGLO vector and then formed the wild type (TUG1-WT, miR-29c-3p-WT, SIRT1-WT) or mutant (TUG1-MUT, miR-29c-3p-MUT, SIRT1-MUT) for co-transfection with mimics NC and miR-29c-3p mimics or TUG1 and vector, respectively. 48 h post-transfection, luciferase activities were measured on Modulus single-tube multimode reader (Promega). And *Renilla* luciferase activity was normalized to firefly luciferase.

### TUNEL assay

HK-2 cells were seeded in 12 well plate containing polylysinecoated slides (Thermo Fisher Scientific, Waltham, MA USA) and cultured for 12 h to allow cell attachment. After transfection and high glucose stimulation, cells were fixed with 4% formaldehyde solution and were subjected to the TUNEL staining using TUNEL assay kit (Roche Molecular Biochemicals, Indianapolis, IN, USA). The fluorescence of HK-2 cells was examined under a fluorescence microscope.

### Statistical analysis

Each experiment was performed at least three times with consistent results, and data were presented as mean ± standard deviation (SD). All statistical analysis was carried out using the SPSS statistical software package (Chicago, IL, USA). Statistical evaluation was performed using Student's *t* test (two-tailed) between two groups or one-way analysis of variance (ANOVA) followed by Tukey post hoc test for multiple comparison. $P < 0.05$ was considered statistically significant.

## Results

### 1. High glucose challenge induces apoptosis, ERS and downregulates TUG1 expression in HK-2 cells

First, we examined the effect of high glucose challenge on the apoptosis of HK-2 cells. The results showed that high glucose induced apparent HK-2 cell apoptosis in a dose-dependent manner as assessed by flow cytometry (Fig 1A). Consistent with the apoptosis-inducing effect of high glucose, we found markedly increased levels of Bax and Caspase3 in high glucose-challenged HK-2 cells. In contrast, the expression of anti-apoptotic Bcl-2 was decreased upon high glucose stimulation (Fig 1B). To explore whether the high glucose stimulation induced apoptosis through ERS pathway, we detected the expression of several marker proteins. And we found that the levels of Caspase12, GRP78, CHOP were observably increased by high glucose stimulation (Fig 1C). Importantly, a significant reduction of lncRNA TUG1 was observed in HK-2 cells treated with high glucose (Fig 1D). These results suggest a potential role of lncRNA TUG1 in regulating high glucose-induced damage of HK-2 cells.

### 2. TUG1 decreases high glucose-triggered apoptosis and ERS in HK-2 cells

In order to explore the modulatory role of TUG1 on high glucose-mediated renal epithelial cell damage, TUG1 was overexpressed in HK-2 cells, and real-time PCR result demonstrated a successful overexpression of TUG1 (Fig 2A). Ectopic expression of TUG1 significantly protected HK-2 cells from apoptosis upon high glucose challenge (Fig 2B). Moreover, the mRNA

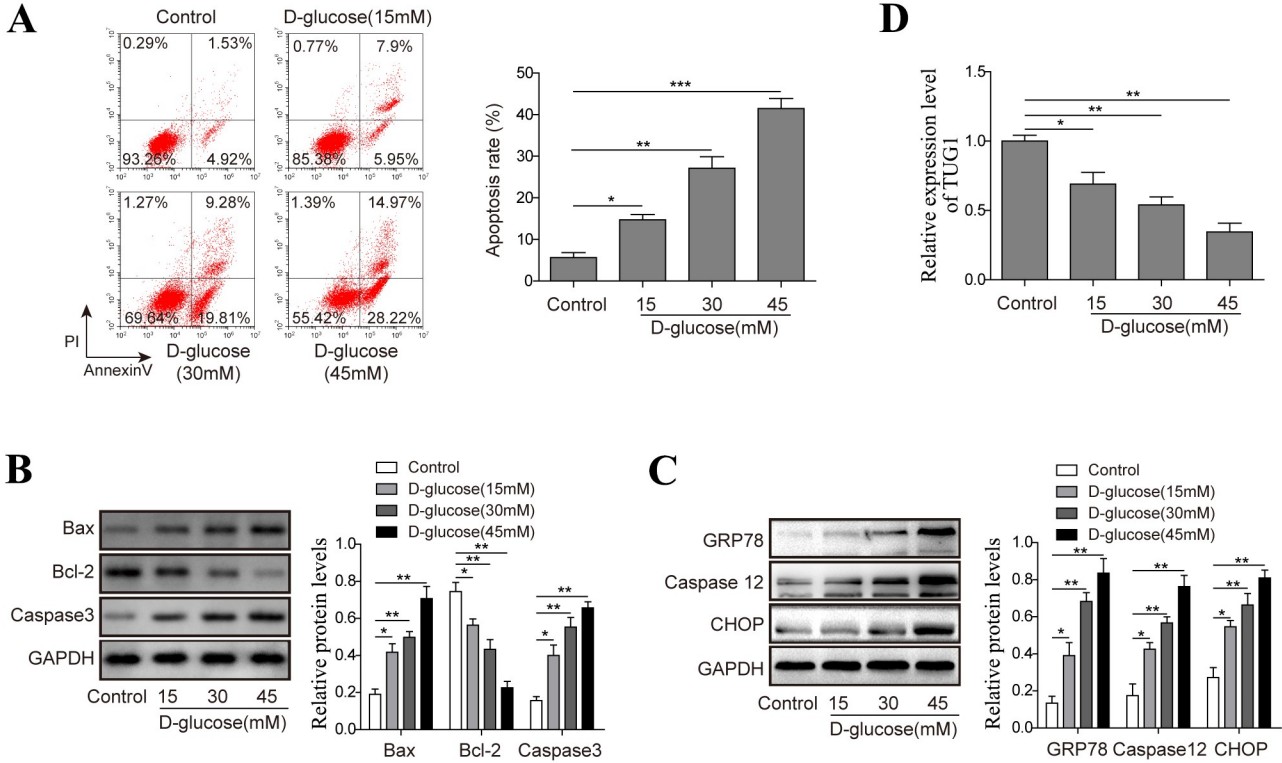

**Fig 1. High glucose (HG) challenge induces apoptosis, ERS and downregulates TUG1 expression in HK-2 cells.** (A) HK-2 cells were treated different concentrations (15, 30, 45 mM) of D-glucose for 48 h, cell apoptosis was examined by flow cytometry. (B) The protein levels of indicated proteins (Bax, Bcl-2, Caspase3) were evaluated by western blot, respectively. (C) The expression of ERS-related proteins (Caspase12, GRP78, CHOP) were measured by western blot. (D) HK-2 cells were treated with 15, 30, 45 mM D-glucose for 48 h, the level of TUG1 was evaluated by real-time PCR. Data were mean ± SD and were representative of three independent experiments. $^{*}p<0.05$, $^{**}p<0.01$, $^{***}p<0.001$.

and protein levels of GRP78, Caspase12 CHOP, p-PERK and p-Eif-2α, which are indicative of high ERS, were significantly diminished by TUG1 overexpression (Fig 2C and 2D, and S1 Fig). Therefore, TUG1 might render HK-2 cells more resistant to high glucose-induced apoptosis by alleviating ERS.

## 3. TUG1 interacts with miR-29c-3p which suppresses SIRT1 expression in HK-2 cells

Next, we verified the interaction among TUG1, miR-29c-3p and SIRT1. From real-time PCR results, the abundance of miR-29c-3p induced by high glucose was significantly reduced by TUG1 overexpression in HK-2 cells (Fig 3A), to a similar extent in normal condition (S2A Fig). By performing sequence alignment in starBase website, we found that TUG1 was putatively interacted with miR-29c-3p (Fig 3B). To further validate this result, we performed dual-luciferase activity assay, and found that miR-29c-3p significantly reduced the luciferase activity in wild type TUG1-expressing HK-2 cells, but not in mutant TUG1-expressing HK-2 cells (Fig 3B). On the other hand, TUG1 also decreased the luciferase activity in miR-29c-3p-WT-expressing cells, but not in mutant miR-29c-3p-expressing cells (Fig 3B). Furthermore, RIP assay revealed the presence of both TUG1 and miR-29c-3p in the same complex as Ago2, the catalytic component in the RNA-induced silencing complex[25] (S3B Fig). Then miR-29c-3p was predicted to interact with SIRT1 (Fig 3C). This prediction was further consolidated by the fact that miR-29c-3p significantly decreased the luciferase activity in HK-2 cells transfected

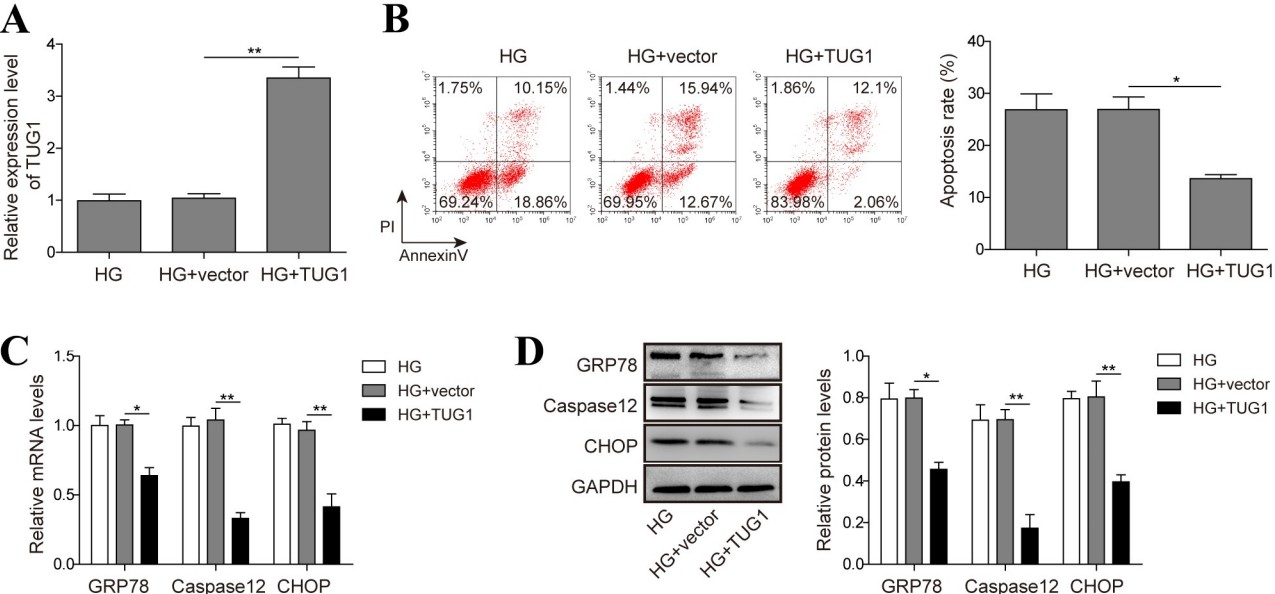

**Fig 2. TUG1 decreases high glucose (HG)-triggered apoptosis and ERS in HK-2 cells.** (A) HK-2 cells were transfected with empty vector or TUG1 overexpression vector, mRNA level of TUG1 was evaluated by real-time PCR. (B) HK-2 cells transfected with indicated vectors were challenged with 30 mM D-glucose for 48 h, cell apoptosis was examined by flow cytometry. (C, D) HK-2 cells were challenged with D-glucose, the levels of GRP78, caspase12 and CHOP were examined by real-time PCR and western blot. Data were mean ± SD and were representative of three independent experiments. *$p < 0.05$, **$p < 0.01$.

with plasmid carrying wild-type SIRT1 3'-UTR, but not in those transfected with plasmid carrying mutant SIRT1 3'-UTR (Fig 3C). Additionally, the downexpression of SIRT1 in high glucose-exposed cells were further decreased by miR-29c-3p (Fig 3D). Thus, TUG1 may function as a sponge for miR-29c-3p, which targets SIRT1 in HK-2 cells.

## 4. MiR-29c-3p exerts pro-apoptotic role in high glucose-challenged HK-2 cells

We then examined the impact of miR-29c-3p on the apoptosis of HK-2 cells. The real-time PCR and western blot results showed that miR-29c-3p inhibitor significantly increased the expression of SIRT1 in HK-2 cells (Fig 4A). By flow cytometry analysis, inhibition of miR-29c-3p significantly protected HK-2 cells from high glucose-induced apoptosis (Fig 4B). Moreover, high glucose-increased levels of ERS-associated proteins (Caspase12, GRP78, CHOP, p-PERK, p-eIF-2α) were down-regulated by miR-29c-3p inhibition in high glucose-challenged HK-2 cells (Fig 4C). Collectively, in contrast to the anti-apoptotic role of TUG1, miR-29c-3p increases the apoptosis of HK-2 cells after high glucose treatment via ERS pathway.

## 5. The protective role of TUG1 is dependent on its ability to downregulate miR-29c-3p expression

Finally, we investigated whether TUG1 modulated the apoptosis of high glucose-challenged HK-2 cells via sponging miR-29c-3p. In a TUNEL-staining assay, forced expression of TUG1 markedly reduced the number of TUNEL[+] apoptotic HK-2 cells, whereas this effected was reversed by miR-29c-3p (Fig 5A and 5B). Consistently, the anti-apoptotic effect of TUG1 was largely abrogated by the concomitant overexpression of miR-29c-3p as assessed by flow cytometry (Fig 5A and 5B). As expected, the expression of pro-apoptotic Bax and Caspase3 was

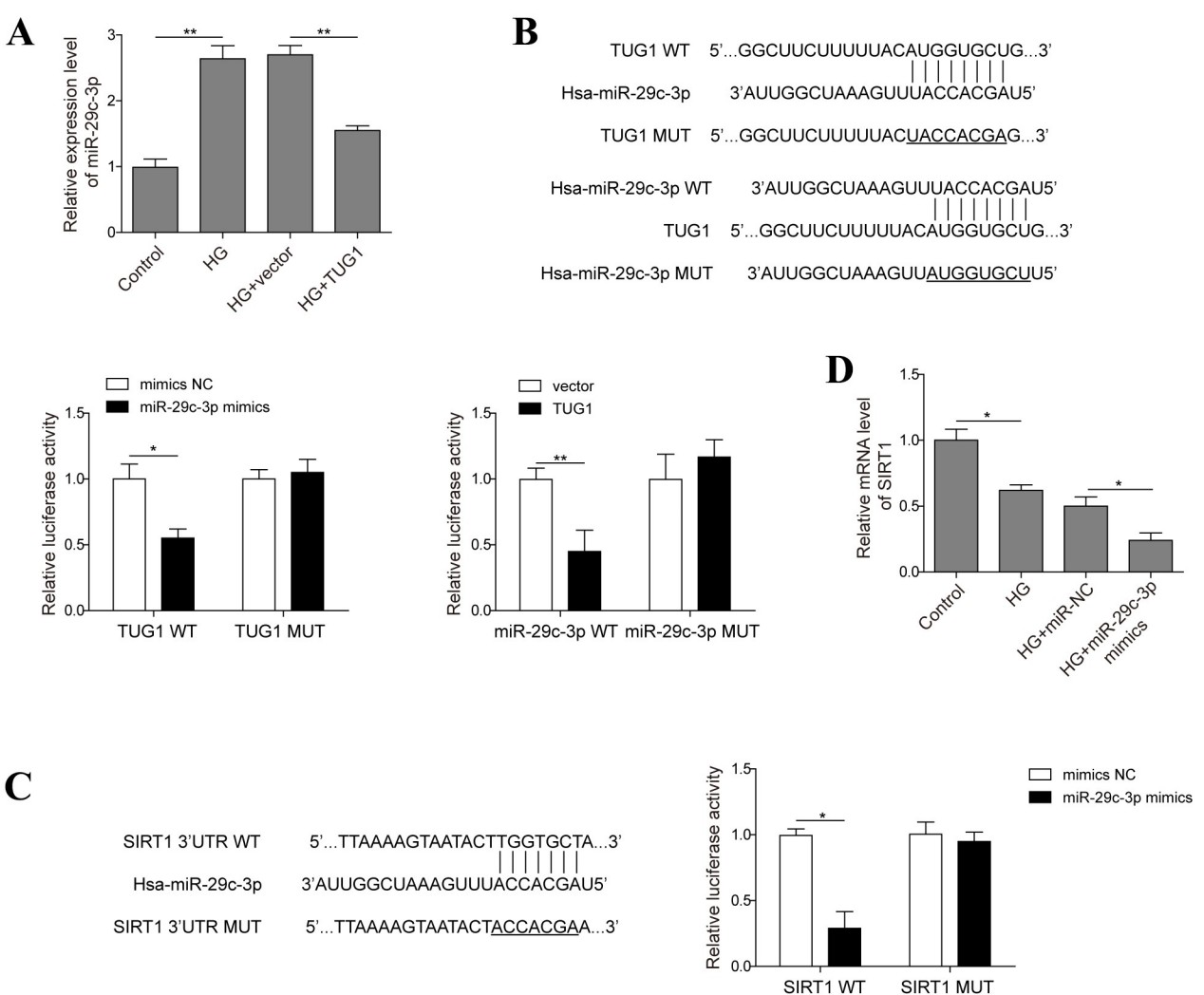

**Fig 3. TUG1 interacts with miR-29c-3p which suppresses SIRT1 expression in HK-2 cells.** (A) HK-2 cells transfected with empty vector or TUG1 overexpression vector were challenged with 30 mM D-glucose for 48 h, mRNA level of miR-29c-3p was evaluated by real-time PCR. (B) Sequence alignment of TUG1 and miR-29c-3p; HK-2 cells were transfected with miR-29c-3p mimics or TUG1 and wild-type/mutant TUG1 or miR-29c-3p respectively, the luciferase activity was examined. (C) Sequence alignment of miR-29c-3p and 3'-UTR region of SIRT1; HK-2 cells were transfected with miR-29c-3p mimics and plasmid carrying wild-type or mutant SIRT1 3'-UTR, the luciferase activity was examined. (D) HK-2 cells were transfected with miR-29c-3p mimics or mimics NC followed by HG incubation, the expression of SIRT1 was evaluated by real-time PCR. Data were mean ± SD and were representative of three independent experiments. $^{*}p<0.05$, $^{**}p<0.01$.

upregulated, whereas the expression of anti-apoptotic Bcl-2 was downregulated in HK-2 cells co-transfected with TUG1 plus miR-29c-3p mimics compared to those transfected with TUG1 alone (Fig 6A). Moreover, miR-29c-3p impaired the ability of TUG1 to decrease the levels of ERS-associated GRP78, caspase12 and CHOP in high glucose-challenged HK-2 cells (Fig 6B). Also, the TUG1-downregulated activation of PERK and eIF-2α was largely reversed by miR-29c-3p (Fig 6C). Thus, TUG1 decreases high glucose-induced ERS and apoptosis by targeting miR-29c-3p for downregulation.

## Discussion

Although most lncRNAs are traditionally considered to be translational noise, many functional lncRNAs have been identified in the past decade. In terms of kidney and cardiovascular

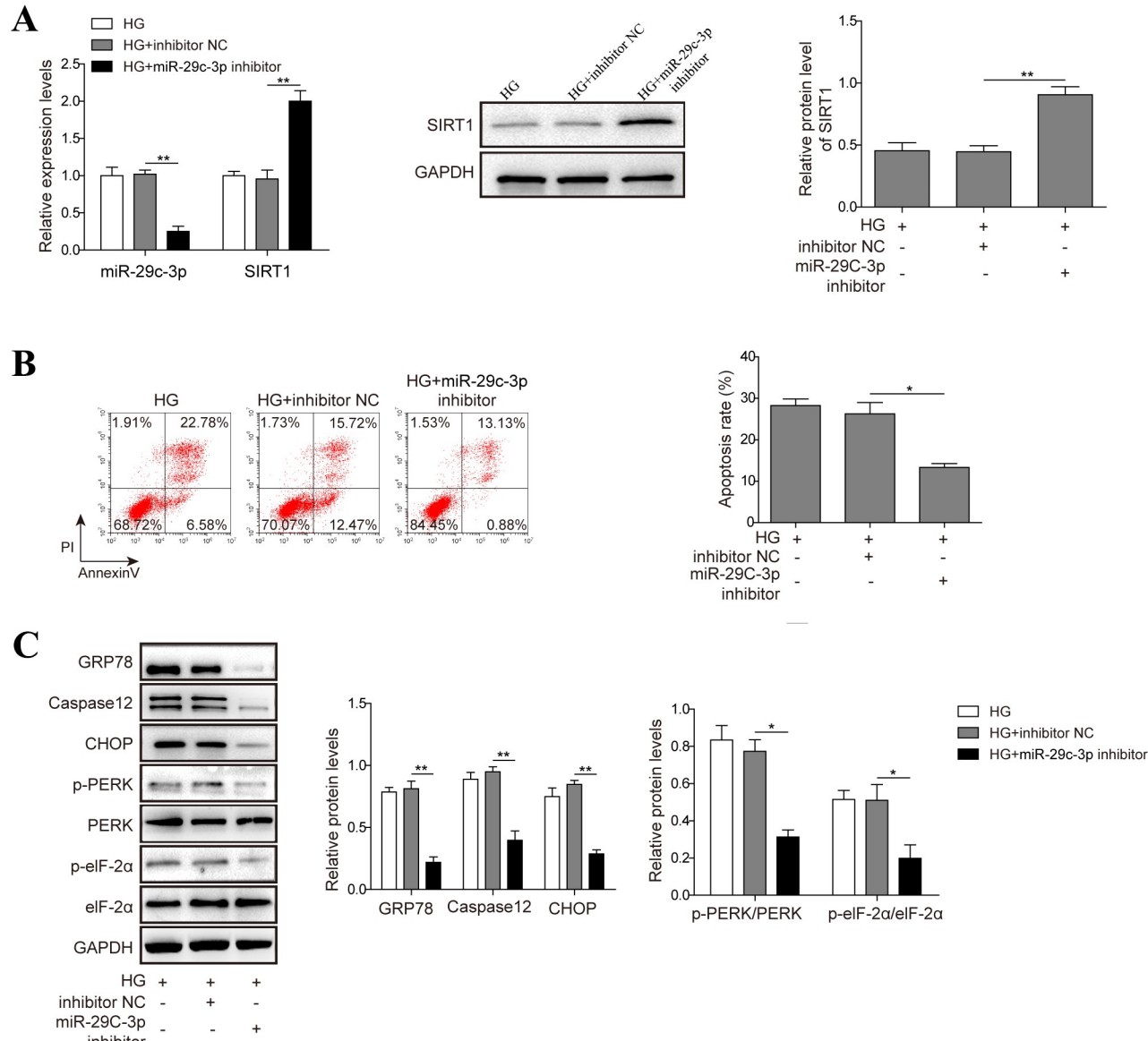

**Fig 4. MiR-29c-3p exerts pro-apoptotic role in high glucose (HG)-challenged HK-2 cells.** (A) HK-2 cells were transfected with miR-29c-3p inhibitor or control inhibitor, the expression of SIRT1 was evaluated by real-time PCR and western blot. (B, C) HK-2 cells were transfected with control inhibitor or miR-29c-3p inhibitor, followed by treatment with D-glucose, cell apoptosis was evaluated by flow cytometry (B), the expression or phosphorylation of indicated proteins (Caspase12, GRP78, CHOP, p-PERK, p-eIF-2α) were examined by western blot (C). Data were mean ± SD and were representative of three independent experiments. $^*p<0.05$, $^{**}p<0.01$.

diseases, Wang et. al. has reported that 1018 lncRNAs had altered expression in mice with diabetic nephropathy[14], but the exact influences of these lncRNA on the pathological processes of diabetic nephropathy were largely unknown.

TUG1 was initially found to be highly expressed in multiple kinds of human cancers, including hepatocellular carcinoma, glioma, oesophageal squamous cell carcinoma and osteosarcoma[26–29]. In these cancers, TUG1 plays oncogenic roles via enhancing the proliferative, migratory capacity of tumor cells. However, in non-small cell lung cancer, TUG1 inhibits the growth of tumor cells and thus serves as a tumor suppressor[30]. Recently, TUG1 was also

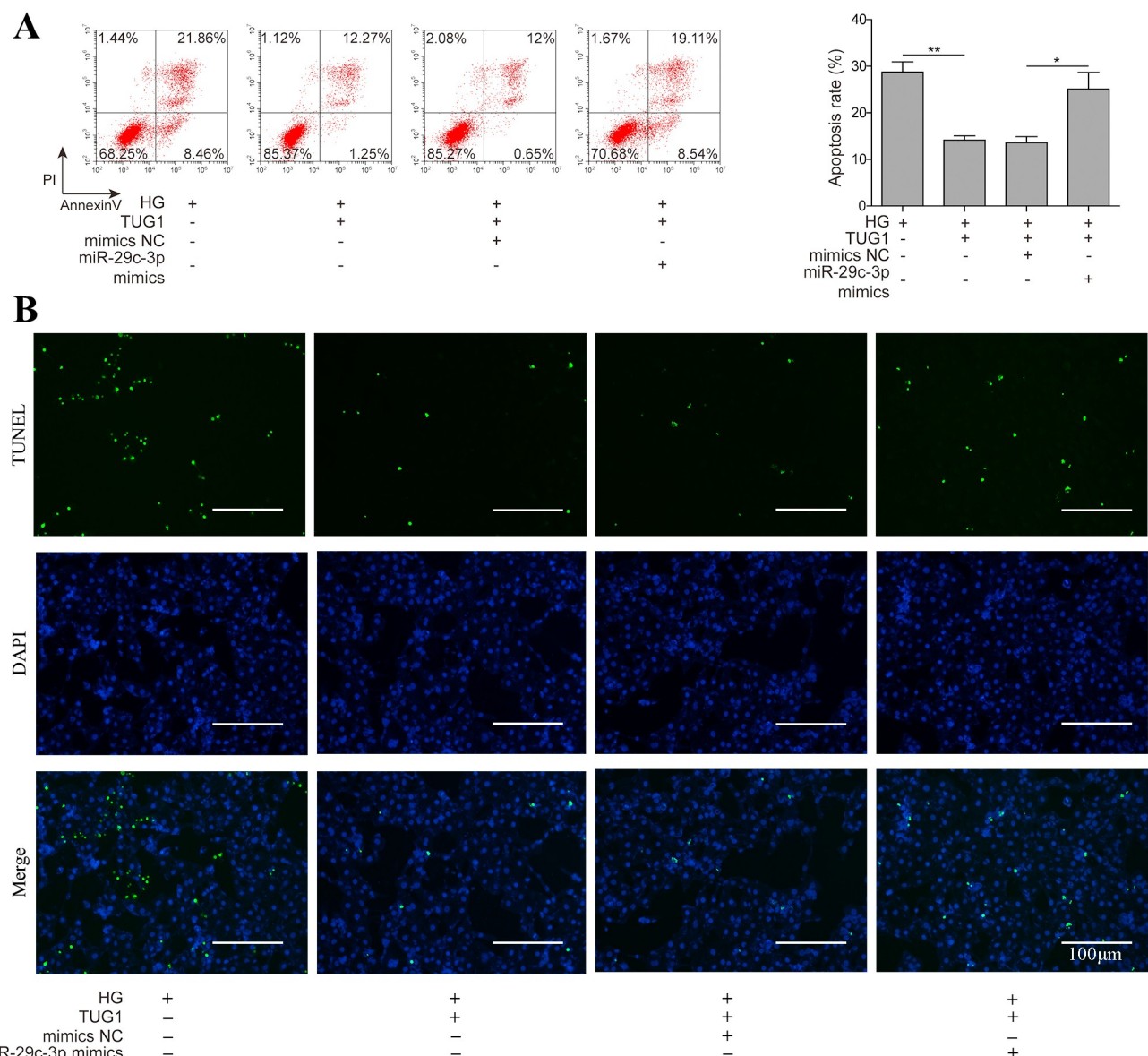

**Fig 5. The protective role of TUG1 is dependent on its ability to downregulate miR-29c-3p expression.** (A, B) HK-2 cells were transfected with TUG1, either alone or in combination with miR-29c-3p mimics, followed by D-glucose treatment for 48 h, cell apoptosis was evaluated by TUNEL staining and flow cytometry.

reported to be a regulator in the pathological processes of diabetic nephropathy by affecting the function of mesangial cells and podocyte cells[16, 31]. TUG1 modulates mitochondrial bioenergetics in diabetic nephropathy[15]. Moreover, TUG1 reduces the accumulation of extracellular matrix accumulation by antagonizing the effect of miRNA-377 in downregulating PPARγ expression in diabetic nephropathy[16]. Also, TUG1 affects the apoptosis of podocytes by modulating pathway in diabetic rats with diabetic nephropathy[31]. On the other hand, miR-29c-3p was reported to modulate the expression of inflammatory cytokines in diabetic nephropathy by suppressing the expression of tristetraprolin[32]. In the present study, we provide a new understanding to this area by revealing that in RTECs, lncRNA TUG1 functions as

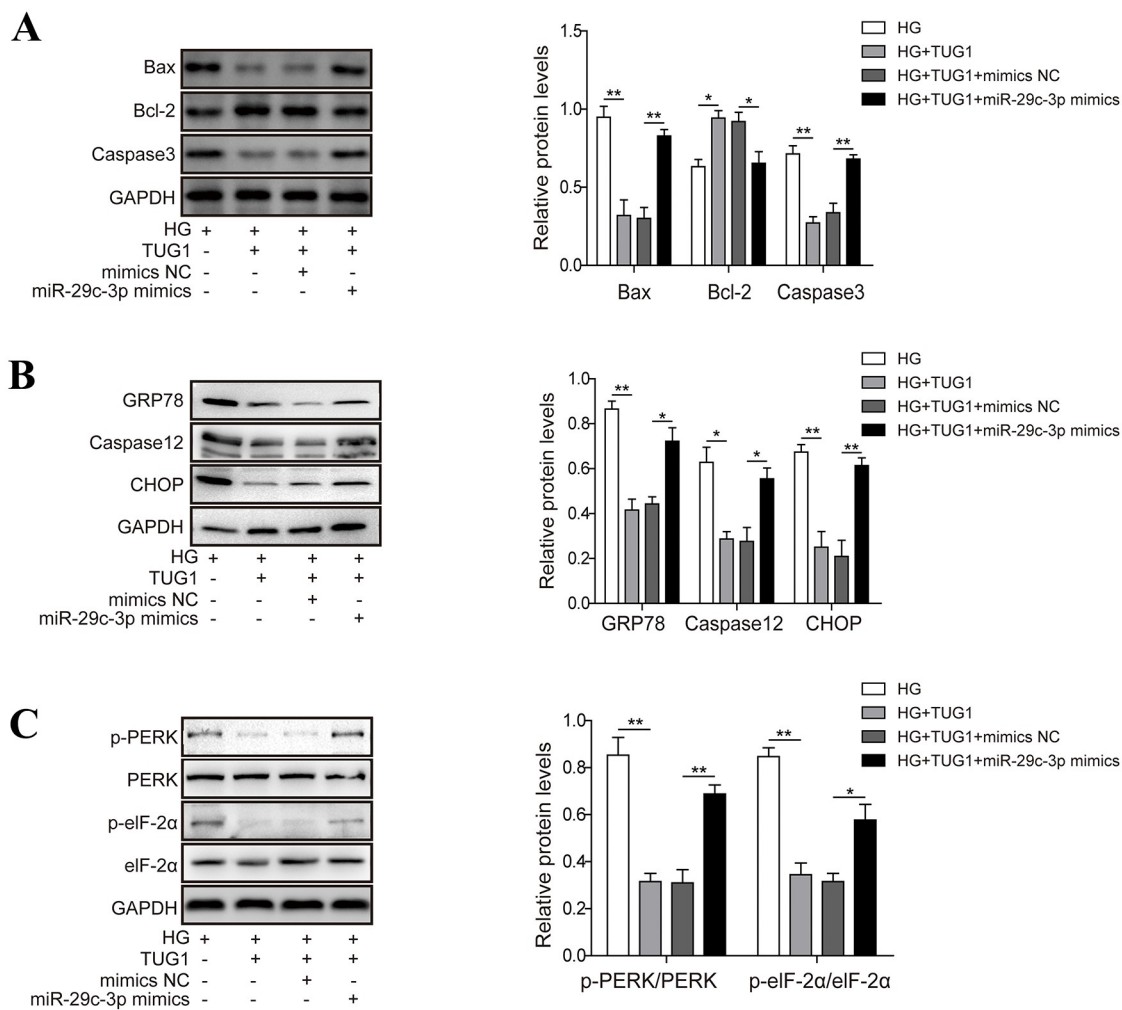

**Fig 6. The protective role of TUG1 is dependent on its ability to downregulate miR-29c-3p expression.** (A-C) HK-2 cells were treated as described in Fig 5, the expression of pro-apoptotic and anti-apoptotic proteins (A), or the expression (B) and phosphorylation (C) of ERS-associated proteins were examined by western blot. Data were mean ± SD and were representative of three independent experiments. $^*p<0.05$, $^{**}p<0.01$.

a suppressor of high glucose-induced ERS and apoptosis through counteracting the effect of miR-29c-3p and thus supporting the expression of SIRT1.

As a nicotinamide adenosine dinucleotide-dependent deacetylase, SIRT1 is well-characterized deacetylase by its ability in modulating cellular glucose metabolism. Many studies have revealed that SIRT1 can mitigate renal disorders via multiple mechanisms, such as reducing oxidative damage, preventing the development of fibrosis, or maintaining mitochondria function[33]. The expression of SIRT1 is reported to be regulated by mircoRNAs. In HK-2 cells, miR-133b and miR-199b targeted SIRT1 for downregulation, leading to the enhanced epithelial to mesenchymal transition and renal fibrosis[34]. In mesangial cells, miR-34a-5p reduced SIRT1 expression and thus aggravated renal fibrosis[35]. Here, we identified another SIRT1-targeing microRNA, miR-29c-3p, whose function can be antagonized by lncRNA TUG1.

A previous study suggested that SIRT1 is able to decrease PERK-eIF-2α signaling pathway and alleviate ERS response in growth-plate chondrocytes[23]. ERS is a common phenomenon in RTECs caused by unfolded protein response (UPR) in the context of diabetic nephropathy

[7, 36]. Although appropriate degree of UPR is helpful for cells to adapt to environmental changes and alleviate cell damage[37]. Persistent and drastic environmental stresses often lead to excessive UPR which causes ERS and impairs the normal structure and function of endoplasmic reticulum, leading to cell apoptosis[37, 38]. The regulation of ERS is particularly important in RTECs because these cells contain abundant endoplasmic reticulum and are often exposed to stress conditions[39]. In this work, high glucose-induced ERS and apoptosis in RTECs was found to be modulated by lncRNA TUG1/miR-29c-3p/SIRT1 axis. Whether TUG1 also participates in regulating ERS induced by other factors, such as free fatty acids or amino acid deprivation, remains further investigation.

In summary, our study uncovers a lncRNA TUG1—miR-29c-3p - SIRT1 network in regulating high glucose-induced apoptosis via ERS in renal epithelial cells, which can hopefully be beneficial to the pharmacological intervention of diabetic nephropathy.

## Supporting information

**S1 Fig. TUG1 decreases high glucose (HG)-triggered p-PERK and p-eIF-2α in HK-2 cells.** (A and B) HK-2 cells were challenged with D-glucose, the levels of p-PERK, and p-Eif-2α were examined by real-time PCR and western blot. Data were mean ± SD and were representative of three independent experiments. $^{**}p < 0.01$.
(TIF)

**S2 Fig. Overexpression of TUG1 significantly elevate the expression of SIRT1.** HK-2 cells transfected with empty vector or TUG1 overexpression vector were challenged with 30 mM D-glucose for 48 h, the expression of SIRT1 was evaluated by western blot.
(TIF)

**S3 Fig. LncRNA TUG1 directly targets the expression of miR-29c-3p.** (A) LncRNA TUG1 downregulates the expression of miR-29c-3p. (B) he interaction of ST7-AS1 or miR-181b-5p with Ago2 from HK-2 cells was examined by RIP assay. Expression levels were examined by real-time PCR. Data were mean ± SD and were representative of three independent experiments. $^{**}p < 0.01$.
(TIF)

**S1 Raw images. The western blots images are provided.**
(PDF)

**S1 File.**
(DOCX)

## Author Contributions

**Conceptualization:** Na Wang, Yingying Zheng.

**Data curation:** Pengfei Yi.

**Methodology:** Na Wang.

**Project administration:** Min Song.

**Supervision:** Wenhui Li.

**Writing – original draft:** Yingying Zheng.

**Writing – review & editing:** Shaoqiang Wang.

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
