## [Decision Letter · Decision Letter 0]

19 Mar 2021

PONE-D-21-01132

LncRNA TUG1/miR-29c-3p/SIRT1 axis regulates endoplasmic reticulum stress-mediated renal epithelial cells injury in diabetic nephropathy model in vitro

PLOS ONE

Dear Dr. Zheng

Thank you for submitting your manuscript to PLOS ONE. After careful consideration, we feel that it has merit but does not fully meet PLOS ONE’s publication criteria as it currently stands. Therefore, we invite you to submit a revised version of the manuscript that addresses the points raised during the review process.

We have now received two referee reports, which are included below. We concur with the referees that the proposed role of LncRNA TUG1 in diabetic nephropathy is potentially interesting even there are few works have been published. One important think I would like to mention that the resolution of figures throughout the manuscript are not very good which need to be improved and both referees also mentioned that. I am sorry that I cannot accept the manuscript in the current form, but we will consider a revised manuscript which is minor revision. I hope that you will find our referees comments helpful.

We look forward to receiving your revised manuscript.

Kind regards,

Abdul Qadir Syed, PhD

Academic Editor

PLOS ONE

Journal Requirements:

Reviewers' comments:

Reviewer's Responses to Questions

**Comments to the Author**

1. Is the manuscript technically sound, and do the data support the conclusions?

Reviewer #1: Yes

Reviewer #2: Yes

2. Has the statistical analysis been performed appropriately and rigorously? 

Reviewer #1: Yes

Reviewer #2: Yes

3. Have the authors made all data underlying the findings in their manuscript fully available?

Reviewer #1: Yes

Reviewer #2: Yes

4. Is the manuscript presented in an intelligible fashion and written in standard English?

Reviewer #1: Yes

Reviewer #2: Yes

5. Review Comments to the Author

Reviewer #1: The manuscript entitled "LncRNA TUG1/miR-29c-3p/SIRT1 axis regulates endoplasmic reticulum stress-mediated renal epithelial cells injury in diabetic nephropathy model in vitro" provides new insights on the role of TUG1 in diabetic nephropathy. This manuscript is well written and sheds new light on the TUG1 role as a suppressor of high glucose-induced endoplasmic reticulum stress. As such, the conclusions are generally supported by the data presented but there are some points which the authors need to be addressed.

Comments

• The author suggests that miR-29c-3p was significantly reduced by TUG1 overexpression. Is TUG1 overexpression decreases miR-29c-3p to a similar extent in normal condition?

• Does SIRT1 level decrease after TUG1 overexpression in high glucose condition (HG + TUG1)?

• In Fig. 2c and 2d, the author should include the p-PERK and pe-EIF2a protein blot.

• The author should present high-resolution IF images.

Reviewer #2: Authors claim that it is a “novel” concept to describe the regulation of SIRT1 level by lncRNA TUG1 and miRNA miR-29c-3p in renal tubular epithelial cells. The relation between TUG1, miR-29c-3p and SIRT1 has been indicated in multiple publications before [for example, Nan Fang Yi Ke Da Xue Xue Bao. 2020 Sep 30;40(9):1325-1331. doi: 10.12122/j.issn.1673-4254.2020.09.16.; Int J Oncol. 2019 Apr;54(4):1317-1326. doi: 10.3892/ijo.2019.4699. Epub 2019 Jan 28. etc]. Therefore, the word “novel” mentioned in the introduction could be eliminated. The role of lncRNA TUG1 and SIRT1 has also been described in literature in terms of renal cells [Biomed Pharmacother. 2018 Aug;104:509-519. doi: 10.1016/j.biopha.2018.05.069.; J Inflamm (Lond). 2021 Mar 4;18(1):12. doi: 10.1186/s12950-021-00278-4.]. A clear summary of known literature about the association of TUG1, miR-29c-3p and SIRT1 needs to be described in introduction.

Abstract section here does not need the description of methods. The reason for focusing particularly on TUG1 and miR-29c-3p to relate to SIRT1 needs to be hinted in abstract.

No therapeutic role of the TUG1, miR-29c-3p and SIRT1 has been shown in the paper, therefore the statement on therapeutic potential should be removed from the abstract.

The entire study is based on HK-2 (human kidney 2) cell line, which is a proximal tubular cell derived from normal kidney. It might be artifactual to describe a process solely based on one cell line. Few other similar cell lines like primary renal proximal tubule epithelial cells (RPTEC) needs to be tested before publication.

Images of the figure panels need to be arranged serially in relation to the text.

Please provide the full western blots as supplement showing the house keeping protein and the subject protein/s in the same blot.

In result section 3, it is claimed that TUG1 interacts with miR-29c-3p by using luciferase assay. Luciferase assay does not prove biochemical interaction. Please provide biochemical evidence of interaction between TUG1 and miR-29c-3p.

TUNEL images are not visible. Please provide high resolution brighter images.

6. PLOS authors have the option to publish the peer review history of their article (what does this mean?). If published, this will include your full peer review and any attached files.

Reviewer #1: No

Reviewer #2: **Yes: **Amitabha Mukhopadhyay

---

## [Author Response · Author response to Decision Letter 0]

6 May 2021

Dear Reviewers:

Thank you for the comments, which are all valuable and very helpful in improving the quality of our manuscript. We have studied your comments carefully and have made corrections which we hope meet your expectation. Revised portions are marked in red in the revised manuscript. Point-by-point responses to your comments are as following.

1. The author suggests that miR-29c-3p was significantly reduced by TUG1 overexpression. Is TUG1 overexpression decreases miR-29c-3p to a similar extent in normal condition?

Response: Thank you for the detailed positive comment. In fact，our previous experiments have verified the interaction between TUG1 and miR-29c-3p in normal condition. In Supplement Fig 3A, RT-qPCR data indicated that overexpression of TUG1 significantly suppressed the expression of miR-29c-3p in HK-2 cells in normal condition.

2. Does SIRT1 level decrease after TUG1 overexpression in high glucose condition (HG + TUG1)? 

Response: Thank you for the valuable suggestion. We agree that it is critical to understand the ceRNA mechanism underlying TUG1/miR-29c-3p/SIRT1 axis.

Accordingly, we performed Western blotting to test that SIRT1 level after TUG1 overexpression. As expected, it was found that overexpression of TUG1 could significantly elevate the expression of SIRT1. The results are shown in Supplement Figure 2.

3. In Fig. 2c and 2d, the author should include the p-PERK and pe-EIF2a protein blot.

Response: We completely agree with this valuable suggestion. Thus, we conducted RT-qPCR and Western blotting to detect the expression level of p-PERK/PERK and p-eIF2α, as shown in Supplement Figure 1 (A and B).

4. The author should present high-resolution IF images.

Response: According to your comment, we have adjusted the resolution of IF images in Figure 5.

Dear Professor Amitabha Mukhopadhyay:

Thank you for the comments, which are all valuable and very helpful in improving the quality of our manuscript. We have studied your comments carefully and have made corrections which we hope meet your expectation. Revised portions are marked in red in the revised manuscript. Point-by-point responses to your comments are as following.

1. Authors claim that it is a “novel” concept to describe the regulation of SIRT1 level by lncRNA TUG1 and miRNA miR-29c-3p in renal tubular epithelial cells. The relation between TUG1, miR-29c-3p and SIRT1 has been indicated in multiple publications before [for example, Nan Fang Yi Ke Da Xue Xue Bao. 2020 Sep 30;40(9):1325-1331. doi: 10.12122/j.issn.1673-4254.2020.09.16.; Int J Oncol. 2019 Apr;54(4):1317-1326. doi: 10.3892/ijo.2019.4699. Epub 2019 Jan 28. etc]. Therefore, the word “novel” mentioned in the introduction could be eliminated.

Response: Thanks for your critical but positive comment. Accordingly, we have carefully read the two articles propose by the reviewer and rechecked the meanings of ‘novel’. we agree that using the word ‘novel’ to descript of the result was confusing. Thus, we have eliminated the word ‘novel’ mentioned in the introduction.

2. The role of lncRNA TUG1 and SIRT1 has also been described in literature in terms of renal cells [Biomed Pharmacother. 2018 Aug;104:509-519. doi: 10.1016/j.biopha.2018.05.069.; J Inflamm (Lond). 2021 Mar 4;18(1):12. doi: 10.1186/s12950-021-00278-4.]. A clear summary of known literature about the association of TUG1, miR-29c-3p and SIRT1 needs to be described in introduction.

Response: We completely agree with this valuable suggestion. Thus, we have carefully read the relevant literature. A clear summary of known literature about the association of TUG1, miR-29c-3p and SIRT1 has been described in introduction.

3. Abstract section here does not need the description of methods. The reason for focusing particularly on TUG1 and miR-29c-3p to relate to SIRT1 needs to be hinted in abstract.

Response: Thank you for the suggestion. This suggestion helps to improve our writing and logic level. In the Abstract section, we have deleted the description of methods and added a description of the correlation between TUG1, miR-29c-3p and SIRT1.

4. No therapeutic role of the TUG1, miR-29c-3p and SIRT1 has been shown in the paper, therefore the statement on therapeutic potential should be removed from the abstract. 

Response: Thank you for the detailed positive comment. Accordingly, we have changed the statement of “target for the prevention” to “promising diagnostic marker” in the abstract.

5. The entire study is based on HK-2 (human kidney 2) cell line, which is a proximal tubular cell derived from normal kidney. It might be artifactual to describe a process solely based on one cell line. Few other similar cell lines like primary renal proximal tubule epithelial cells (RPTEC) needs to be tested before publication. 

Response: Thank you for the valuable suggestion. We agree that it might be artifactual to describe a process solely based on one cell line. The experiment data obtained with one cell line (HK-2) was limited and is critical to need another cell line to be test.

 In fact, two cell lines (HK-2 and RPTEC) were selected in the project (Nursery research project of the Affiliated Hospital of Jining Medical University; No. MP-2018-001) design stage. But when applying for the National Natural Science Foundation of China, one of the reviewers pointed out “It has been reported that LncRNA TUG1s can improve the pathological changes of diabetic nephropathy by strengthening the mitochondrial function of podocytes. MiR-29c-3p is also a mature miRNA in renal fibrosis. The study explored the molecular signal pathway only in HK-2 and RPTEC cell lines. Since the mice with LncRNA TUG1s KO or KI can be constructed, the primary podocytes, endothelial cells and mesangial cells can be isolated and studied, which is closer to the pathological state in vivo, and we can further explore which kind of cells and in which way LncRNA TUG1s has more obvious effect on ERS”. We adopted the reviewer’s opinion and recently commissioned Gempharmatech Co., Ltd to design TUG1 KO mice. In view of insufficient funding, we are sorry that we did not use RPTEC for research.

6. Images of the figure panels need to be arranged serially in relation to the text.

Response: Thank you very much for the suggestions. According to your comment, we have rearranged the figure panels.

7. Please provide the full western blots as supplement showing the house keeping protein and the subject protein/s in the same blot.

Response: We agree with the reviewer that it is helpful to demonstrate the entire Western blot. However, after electrophoresis and transfer onto PVDF membrane, we cut the membrane, according to the marker protein, into small pieces containing the pre-detected proteins. Each strip was individually imaged using BIO-RAD ChemiDocTM MP Imaging System, as shown in the following figure. Thus, we apologize not to be able to provide the full size of the Western blot. We are convinced of the authenticity of the experimental western blots.

8. In result section 3, it is claimed that TUG1 interacts with miR-29c-3p by using luciferase assay. Luciferase assay does not prove biochemical interaction. Please provide biochemical evidence of interaction between TUG1 and miR-29c-3p.

Response: Thank you very much. This comment is valuable and helpful for revising and improving our paper. According to your suggestion, we have designed and performed RIP experiment to evaluate the biochemical evidence of interaction between TUG1 and miR-29c-3p. The results were shown in Supplement Figure 3B.

9. TUNEL images are not visible. Please provide high resolution brighter images.

Response: Thank you for the valuable suggestion. We reformatted Figure 5 and have provided high resolution brighter image.

---

## [Decision Letter · Decision Letter 1]

24 May 2021

LncRNA TUG1/miR-29c-3p/SIRT1 axis regulates endoplasmic reticulum stress-mediated renal epithelial cells injury in diabetic nephropathy model in vitro

PONE-D-21-01132R1

Dear Dr. Zheng,

We’re pleased to inform you that your manuscript has been judged scientifically suitable for publication and will be formally accepted for publication once it meets all outstanding technical requirements.

Kind regards,

Abdul Qadir Syed, PhD

Academic Editor

PLOS ONE

**Comments to the Author**

1. If the authors have adequately addressed your comments raised in a previous round of review and you feel that this manuscript is now acceptable for publication, you may indicate that here to bypass the “Comments to the Author” section, enter your conflict of interest statement in the “Confidential to Editor” section, and submit your "Accept" recommendation.

Reviewer #1: All comments have been addressed

Reviewer #2: All comments have been addressed

2. Is the manuscript technically sound, and do the data support the conclusions?

Reviewer #1: Partly

Reviewer #2: Yes

3. Has the statistical analysis been performed appropriately and rigorously? 

Reviewer #1: Yes

Reviewer #2: I Don't Know

4. Have the authors made all data underlying the findings in their manuscript fully available?

Reviewer #1: Yes

Reviewer #2: Yes

5. Is the manuscript presented in an intelligible fashion and written in standard English?

Reviewer #1: Yes

Reviewer #2: Yes

6. Review Comments to the Author

Reviewer #1: (No Response)

Reviewer #2: Dear Editor,

Authors have addressed to all the comments, added new data, reformatted the entire manuscript. I would recommend to accept it for publication.

Thank you.

Amitabha

7. PLOS authors have the option to publish the peer review history of their article (what does this mean?). If published, this will include your full peer review and any attached files.

Reviewer #1: No

Reviewer #2: **Yes: **Amitabha Mukhopadhyay

---

## [Editor Report · Acceptance letter]

27 May 2021

PONE-D-21-01132R1 

LncRNA TUG1/miR-29c-3p/SIRT1 axis regulates endoplasmic reticulum stress-mediated renal epithelial cells injury in diabetic nephropathy model *in vitro*

Dear Dr. Zheng:

I'm pleased to inform you that your manuscript has been deemed suitable for publication in PLOS ONE. Congratulations! Your manuscript is now with our production department. 

Kind regards, 

on behalf of

Dr. Abdul Qadir Syed 

Academic Editor

PLOS ONE